# Linking Immunity with Genomics in Sarcomas: Is Genomic Complexity an Immunogenic Trigger?

**DOI:** 10.3390/biomedicines9081048

**Published:** 2021-08-19

**Authors:** Siddh van Oost, Debora M. Meijer, Marieke L. Kuijjer, Judith V. M. G. Bovée, Noel F. C. C. de Miranda

**Affiliations:** 1Department of Pathology, Leiden University Medical Center, 2333 ZA Leiden, The Netherlands; S.van_Oost@lumc.nl (S.v.O.); D.M.Meijer@lumc.nl (D.M.M.); marieke.kuijjer@ncmm.uio.no (M.L.K.); N.F.de_Miranda@lumc.nl (N.F.C.C.d.M.); 2Centre for Molecular Medicine Norway (NCMM), Faculty of Medicine, University of Oslo, 0318 Oslo, Norway

**Keywords:** sarcoma, genomics, heterogeneity, checkpoint blockade, tumor microenvironment, immunotherapy

## Abstract

Sarcomas comprise a collection of highly heterogeneous malignancies that can be grossly grouped in the categories of sarcomas with simple or complex genomes. Since the outcome for most sarcoma patients has barely improved in the last decades, there is an urgent need for improved therapies. Immunotherapy, and especially T cell checkpoint blockade, has recently been a game-changer in cancer therapy as it produced significant and durable treatment responses in several cancer types. Currently, only a small fraction of sarcoma patients benefit from immunotherapy, supposedly due to a general lack of somatically mutated antigens (neoantigens) and spontaneous T cell immunity in most cancers. However, genomic events resulting from chromosomal instability are frequent in sarcomas with complex genomes and could drive immunity in those tumors. Improving our understanding of the mechanisms that shape the immune landscape of sarcomas will be crucial to overcoming the current challenges of sarcoma immunotherapy. This review focuses on what is currently known about the tumor microenvironment in sarcomas and how this relates to their genomic features. Moreover, we discuss novel therapeutic strategies that leverage the tumor microenvironment to increase the clinical efficacy of immunotherapy, and which could provide new avenues for the treatment of sarcomas.

## 1. Introduction

Sarcomas are a heterogeneous group of tumors arising in the bone and soft tissue. Currently, the World Health Organization recognizes over 70 distinct sarcoma subtypes, which illustrates the biological complexity of these tumors [1]. This complexity also implies considerable challenges regarding their diagnosis and treatment. Conventional treatments, such as surgical resection, chemotherapy, and radiotherapy, are the mainstay of treatment, but the survival of patients has barely improved over the last decades. Therefore, there is an urgent need for effective treatment options. In recent years, immunotherapy, mainly through the advent of T cell checkpoint blockade therapies, has revolutionized the treatment of a number of solid cancers, particularly ones with immunogenic features. In sarcomas, however, it currently has limited use as only a small group of patients benefits from these therapies. Nevertheless, recent studies have highlighted immunogenic features in some sarcomas that may support immunotherapeutic approaches for their treatment.

At the genomic level, a simplified distinction can be made between sarcomas with simple or complex genomes. Sarcomas with simple genomes often harbor a recurrent driver genomic event (e.g., translocation, mutation, or amplification), while other extensive alterations throughout the genome are not observed (Figure 1A). For instance, Ewing sarcoma contains a gene fusion between *EWSR1* (or *FUS*) and a member of the ETS family of transcription factors (e.g., *EWSR1-FLI1*) [2]. Next to a few secondary mutations in *TP53* and *STAG2*, Ewing sarcomas generally lack other genetic alterations. In addition, well-differentiated liposarcoma and dedifferentiated liposarcoma are associated with amplifications in a specific region on chromosome 12 (12q13-15) that encompasses genes such as *MDM2* and *CDK4* [3]. Other well-known examples of sarcomas with simple genomes include alveolar soft-part sarcoma and synovial sarcoma. In contrast, sarcoma subtypes with complex genomes present extensive, non-recurrent genetic alterations, including single nucleotide variants (SNVs) and structural variants (Figure 1B) [4]. Mutational patterns reminiscent of catastrophic genomic events, such as *chromothripsis* and *kataegis,* are commonly observed in tumors with complex genomes, such as osteosarcomas [5]. Since these genomic alterations are so extensive and largely non-recurrent, sarcomas with complex genomes are highly heterogeneous entities, both intratumorally as well as across patients. This has complicated the understanding of the mechanisms underlying tumorigenesis in these cancers [6]. Other sarcomas with complex genomes include myxofibrosarcoma, undifferentiated soft tissue sarcoma, and leiomyosarcoma. Of note, sarcomas with simple genomes may progress towards complex genomes through the acquisition of secondary mutations and chromosomal instability, as seen in low-grade chondrosarcomas carrying an *IDH1* or *IDH2* mutation in a simple genomic background that develop into high-grade chondrosarcomas with complex genomes [7].

Genetic alterations can be perceived by the immune system via various mechanisms. Mutations in the coding genome that impact protein sequences can trigger T cell immunity when neoepitopes undergo antigen presentation in the context of the Human Leukocyte Antigen (HLA) system [8]. In addition, DNA sensing pathways can be activated by the presence of cytosolic DNA as a result, for instance, of faulty mitotic events [9]. In response, signaling cascades that lead to the production of pro-inflammatory cytokines are triggered, thereby attracting immune cells to the tumor microenvironment. The link between antigen presentation and response to T cell checkpoint blockade immunotherapies is well-established, as demonstrated by the exceptional responses to this type of immunotherapy in cancers with high mutation burden, such as non-small cell lung cancer and melanoma, as well as mismatch repair deficient (MMRd) cancers [10]. In sarcomas with complex genomes, such as osteosarcoma, the tumor mutation burden is generally low (~1.15 mutations per Mb) [11]. Chromosomal instability, on the other hand, is frequently observed in complex sarcomas and could promote immune responses via sensing of cytosolic DNA by the cyclic GMP-AMP synthase-Stimulator of Interferon Genes (cGAS-STING) pathway or through upregulation of ligands that can activate effector immune cells, such as NK cells [12,13]. Hence, it is hypothesized that sarcomas with complex genomes would have an increased probability of benefiting from immunotherapy as compared to sarcomas with simple genomes. Conversely, chromosomal aberrations can also promote immune evasion and reduced response to immunotherapy [14]. Although infrequent, responses to immunotherapy have been reported in sarcoma patients, which advocates further elucidation of their biology, including the role of genetics and the tumor microenvironment (TME) in underlying responses to immunotherapy [15].

## 2. The Tumor Microenvironment in the Context of Genomic Complexity

The composition and functional orientation of the cells that compose the TME, including immune cells, stromal cells, mesenchymal stem cells, endothelial cells, and pericytes, play a fundamental role in determining cancer progression and response to (immuno-)therapy. In particular, the TME can exhibit features that are indicative of ongoing antitumor immunity, as indicated by the presence of cytotoxic T cells and pro-inflammatory signals, or display hallmarks of immune suppression, such as the ones related to the activity of immune inhibitory pathways (e.g., IL-10, TGF-β) [16,17]. A TME enriched with lymphocytic infiltration, and pro-inflammatory signals are often referred to as immunologically “hot”. However, cancers often present with immune-suppressed TMEs that can either be classified as immune-excluded or immunologically “cold” [18]. In tumors with an immune-excluded phenotype, cytotoxic T lymphocytes (CTLs) are confined to the periphery of tumors, barred from infiltrating the tumor core by immune suppressive signals or physical barriers provided by cancer, immune, and stromal cells as well as by an aberrant extracellular matrix. Immunologically “cold” tumors are generally defined by the overall absence of CTLs either due to a lack of immunogenic features in cancer cells or as a result of various immunosuppressive mechanisms. Importantly, non-cancer cells encompassed in the TME can also aid in immune evasion through various mechanisms of immunosuppression that, for instance, include the expression of immune suppressive molecules [19,20]. Impairment of the antigen-presenting machinery in tumor cells, for instance, through defective HLA class I expression, is another prominent means of immune escape [21,22].

While sarcomas generally present with an immunologically “cold” TME [23], immunogenic features can be encountered in a relevant proportion of cases [24]. The fact that predominantly sarcomas with complex genomes present immunogenic features indicate an association between genome complexity and immune response in these tumors. An overview of the TME compositions for several example sarcoma types is given in Table 1 to illustrate the diversity in terms of genomics and TME.

### 2.1. The Tumor Microenvironment of Sarcomas with Simple Genomes

Sarcomas with simple genomes, such as Ewing sarcomas, synovial sarcomas, and alveolar soft-part sarcomas, harbor isolated genomic alterations and present with a low mutation burden. Accordingly, they are, in general, poorly infiltrated by immune cells. In addition to having a low tumor mutation burden, Ewing sarcomas are, in many cases, noted to have a low expression of Human Leukocyte Antigen (HLA) class I and to have acquired expression of the immunosuppressive HLA-G molecule at the surface of tumor cells. Both observations have previously been associated with decreased levels of CD8+ T cell infiltration [54,55]. Even though lymphocytes are usually absent in Ewing sarcomas, most cases do contain tumor-associated macrophages (TAMs) with anti-inflammatory properties as well as immunosuppressive monocytes [36,39]. The presence of these cells might be an important contribution to the lack of lymphocytic infiltration in Ewing sarcomas. Nevertheless, some exceptional cases show moderate levels of infiltration by T cells, which has been associated with interferon-γ signaling [56]. Similar observations have been made in synovial sarcomas and alveolar soft-part sarcomas, as these malignancies often encompass myeloid cells with immunosuppressive traits and, only occasionally, display moderate levels of lymphocytes co-occurring with HLA class I expression in tumor cells [25,38,57,58].

Regarding immunotherapeutic targets, most sarcomas with simple genomes show little to no expression of PD-L1 or infiltration of T cells expressing PD-1 or cytotoxic T lymphocyte antigen 4 (CTLA-4) (Table 1) [30,32,44,59,60]. Surprisingly, PD-L1 expression has been observed in alveolar soft-part sarcoma patients of which the majority had received either chemo-, molecular targeted-, or immuno-therapy, or multiple lines of treatment prior to this particular study [26]. This group of tumors was infiltrated with T lymphocytes as well, which raises the question of whether T cell infiltration and PD-L1 expression had been induced by previous treatments.

Although well-differentiated liposarcomas are rarely infiltrated by T cells, roughly 50% of cases feature tertiary lymphoid structures (TLSs) in their TME [31]. TLSs are newly formed lymphoid structures that develop at sites of chronic inflammation, such as in a pro-inflammatory TME. They are comprised of T cells, B cells, and dendritic cells, resembling a germinal center structure in a lymph node [61]. Moreover, TLSs displaying PD-L1 expression are found in both fusion-driven alveolar rhabdomyosarcoma and non-fusion-driven embryonal rhabdomyosarcoma [62]. What triggers the formation of these TLSs in well-differentiated liposarcomas, and rhabdomyosarcomas is still unknown but warrants further elucidation as these structures have been associated with good response to immunotherapy in sarcomas. In addition, CD8+ T cells infiltrated into well-differentiated liposarcomas can express PD-1, suggesting those could be harnessed by therapies targeting the PD-1/PD-L1 axis [31].

Sarcomas with simple genomes are often driven by translocations, many of which involve transcription factors. The fusion product can profoundly reshape the transcriptional profiles of these cancers, potentially leading to the de novo expression of immunomodulatory molecules in the TME. Indeed, this has been suggested to occur in alveolar soft-part sarcoma, since TFE3, the transcription factor translocated and overexpressed in these tumors, is involved in the regulation of expression of transforming growth factor-beta (TGF-β), a well-known mediator of immune suppression, as well as the receptor tyrosine kinase MET, which can affect PD-L1 expression through activation of the PI3K pathway [63,64].

### 2.2. The Tumor Microenvironment in Sarcomas with Complex Genomes

The observation that, in general, sarcomas with complex genomes are more often infiltrated by effector immune cells suggests a connection between chromosomal instability and immunogenicity. A pan-cancer analysis showed that CTL infiltration is independent of the number of expressed neoantigens in cancer types that are characterized by copy number alterations, including sarcomas [65]. Furthermore, using multi-omics analyses, the authors of that study found an association between CTL infiltration in cancers with chromosomal instability and phosphorylation of the Ataxia Telangiectasia Mutated (ATM) protein, a DNA double-strand break damage response protein. They showed that ATM phosphorylation levels were positively correlated with the expression of CCL5, CXCL10, and IL-16. These chemokines and cytokines are known T cell attractants, indicating that chromosomal instability may modulate CTL infiltration via ATM signaling. Interestingly, *ATM* is frequently mutated or affected by copy number alterations in sarcomas with complex genomes, including myxofibrosarcoma and leiomyosarcoma, which could possibly affect the extent of immune cell infiltration in those [66,67]. As discussed, chromosomal instability can also lead to the release of genomic DNA into the cytosol of cells and to the subsequent activation of DNA-sensing pathways, such as the cGAS-STING pathway, which, in turn, can evoke an antitumor immune response. While the cGAS-STING mechanism has been extensively studied in other tumors [68], not much is known about its activity in sarcomas. Of note, the activity of this pathway can be silenced during tumorigenesis, as demonstrated in melanomas, where epigenetic silencing of STING pathway genes can occur [69]. Altogether, in sarcomas, it is likely that chromosomal instability contributes to a more immunogenic TME, but additional research will be necessary to underpin a mechanistic link.

Even though genetically complex sarcomas harbor many genomic alterations, most subtypes are considered to be immunologically “cold” (e.g., osteosarcoma, chondrosarcoma, and leiomyosarcoma) (Table 1). Interestingly, in osteosarcoma, an association has been reported between genomic instability and immunogenicity. Wang et al. (2019) showed a higher occurrence of chromosomal instability and putative neoantigens in metastatic osteosarcoma compared to matched primary tumors [70]. In parallel, they also demonstrated that metastatic osteosarcomas more frequently contained higher numbers of tumor cells expressing PD-L1 and were infiltrated by more T cells. Comparable results were also found by another study that reported significantly higher numbers of TILs and of PD-L1 expression in osteosarcoma metastases compared to primary tumors [71]. However, contradictory results were found by Wu et al. (2020), where no significant differences were found between primary, recurrent, and metastatic osteosarcoma in the number of somatic alterations and potential neoantigens [72]. Of note, the extent of immune infiltration could also be related to the location of the metastatic lesion. For example, osteosarcoma commonly metastasizes to lung tissue, which has different tissue characteristics compared to the primary tumors originating from bone. This possibly affects the accessibility of immune cells to the tumor tissue.

In recent years, several studies have identified soft tissue sarcomas with an immunologically “hot” TME, including subsets of dedifferentiated liposarcomas, undifferentiated soft tissue sarcomas, and myxofibrosarcomas (Table 1). By integrating publicly available sarcoma datasets, Petitprez et al. (2020) identified five sarcoma immune classes, two of which being immunologically “hot” and encompassing a total of 33.3% of the cases [24]. One of these immune “hot” subsets, comprising 17.8% of all cases, was found to be not only enriched with T lymphocytes, myeloid cells, and immune checkpoint expression but also with B cells and TLSs. Concurrently, the authors observed a relatively low mutational burden across all sarcoma immune classes, which suggests that other mechanisms are driving the immunologically “hot” microenvironment in both classes. Indeed, another study that utilized the same dataset highlighted a role for chemotaxis, interferon-γ signaling, and antigen presentation in the shaping of the pro-inflammatory TME phenotype in these sarcomas [73]. Nevertheless, soft tissue sarcomas with a high mutational burden have been observed, including angiosarcomas and undifferentiated soft tissue sarcomas [74,75,76]. Interestingly, these tumors were superficially located and contained UV-related mutational signatures similar to melanomas. In concordance with their high mutational load (~21–68 mutations per Mb), UV-exposed angiosarcomas and undifferentiated soft tissue sarcomas were found to be enriched with pro-inflammatory and immune-related (transcriptional) signatures [75,76,77].

While several sarcoma types with complex genomes are often considered “cold”, the TME composition within specific sarcoma types can be quite diverse, resulting in subsets displaying a “hot” phenotype (Table 1). This may be linked to the stochastic and heterogeneous nature of the genomic alterations that occur in these tumors. For example, the mutational landscape of uterine leiomyosarcomas shows highly heterogeneous patterns of chromosomal alterations [78,79]. A study into the adaptive and innate immune cell landscape of uterine leiomyosarcomas showed that half of the cases were classified as immune cold. However, infiltrate represented by TAMs, T cells, and NK cells was seen in the other half of the cases [52]. Unfortunately, studies correlating somatic chromosomal alterations to the amount of immune infiltrate and immune composition in uterine leiomyosarcomas are lacking. The investigation of a potential association between the extent of genomic alterations and immune infiltrates in leiomyosarcoma would be of great interest.

Chordomas—a rare subtype of sarcoma of notochordal origin, which arise either in the skull base, the spine, or in the sacrum [80]—are, potentially the most immunologically “hot” sarcoma type. This is exemplified by their frequent infiltration with considerable amounts of CD4+ and CD8+ T cells, as well as with M1-like and M2-like TAMs [39,40,41,42]. In addition, at the protein level, PD-L1 is frequently expressed on tumor cells, whereas HLA class I is often expressed at varying degrees within a lesion [81]. These findings further support an immunogenic character of chordomas. It is remarkable that conventional chordomas are so highly infiltrated as their mutational burden is relatively low. Genetic features of chordomas include chromosomal copy number loss of the tumor suppressor gene *CDKN2A* as well as structural variants in genes that encode members of the chromatin-remodeling complex, including *PBRM1* and *SMARCB1* [82,83]. Of note, loss-of-function mutations in these two genes were found to be poor prognostic factors in chordomas, indicating a pivotal role for epigenetic deregulation in progression of chordomas. Interestingly, mutations affecting these chromatin-remodeling genes have also been associated with immunogenic features of several solid cancers. For instance, mutations in *PBRM1* associate with increased CTL infiltration and PD-L1 expression, as well as with decreased infiltration by regulatory T cells in clear cell renal cell carcinoma [84,85]. In addition, mutations in *PBRM1* were also found predictive for worse clinical outcomes after PD-L1 blockade in various cancers, including clear cell renal cell carcinoma and lung adenocarcinoma, highlighting the ambiguous role of *PBRM1* in immunity [86]. In chordomas, however, these findings suggest that alterations in the chromatin-remodeling complex, in part, could explain the extensive infiltration. As epigenetic deregulation through genetic alterations frequently appears in chordomas, it would be interesting to study its role in relation to immunogenicity further.

## 3. Clinical Responses to Immunotherapy in Relation to the Immunogenomics of Sarcomas

### 3.1. Response to T Cell Checkpoint Blockade

Over the years, immunotherapy with T cell checkpoint blockade antibodies has proven to be an excellent strategy to treat a subset of cancer patients [87]. In particular, this approach has been effective in immunogenic cancers that present an immunologically “hot” microenvironment [18]. On the other hand, cancers that do not elicit robust, spontaneous antitumor immune responses are, in general, poor candidates to benefit from checkpoint blockade therapies. Currently, several biomarkers exploiting features related to immunogenicity are being applied to guide patient selection for immune checkpoint inhibition, including tumor mutation burden, mismatch repair-deficiency, and PD-L1 expression. These features have been predictive for response to checkpoint blockade in cancers such as melanoma, non-small cell lung cancer, or colon cancer [10]. Thus, many clinical trials with T cell checkpoint blockade have enrolled sarcoma patients using these markers for selection, of which an all-encompassing overview is presented in the recent review article by Chew and colleagues [88]. In general, sarcomas have a relatively low mutational burden, infrequently express PD-L1, and less than 2% display defects in the DNA mismatch repair system [89,90,91]. Despite the absence of these predictive biomarkers, a considerable fraction of sarcoma patients respond to checkpoint blockade which supports the pursue of immunotherapy in sarcoma. However, this observation also highlights that good predictive biomarkers for response to immune checkpoint inhibition are still lacking, particularly in sarcoma, emphasizing the need to improve our understanding of the underlying biology of this disease.

Alveolar soft-part sarcomas are an exceptional sarcoma subtype in relation to checkpoint blockade, as approximately half of all patients with this disease are responsive to this immunotherapy although the mechanisms of response are still elusive. Given their immunologically “cold” microenvironment, lymphocytic infiltration is often absent. PD-L1 expression has been reported in 50–100% of alveolar soft-part sarcoma but did not correlate with clinical response to PD-1 blockade [26,92]. Moreover, alveolar soft-part sarcomas do not harbor many mutations aside from their characteristic *TFE3-ASPCR1* fusion. Interestingly, it was suggested that few cases of alveolar-soft part sarcomas that responded well to checkpoint blockade exhibited MMRd [92,93], although the alleged prevalence of MMRd in alveolar soft-part sarcomas could not be confirmed in a larger cohort [91]. As opposed to tumors with well-known immunogenic features, such as a high mutation burden, it is speculated that the specific fusion found in alveolar soft-part sarcomas influences immune-related pathways underlying response to anti-PD-1 treatment. Apart from alveolar soft-part sarcomas, no remarkable responses to checkpoint blockade have been observed in the remaining sarcomas with a simple genome.

Even though sarcomas with complex genomes contain higher numbers of immune cells, in general, they also respond poorly to immunotherapeutic agents. Treatment with pembrolizumab, which targets PD-1, led to a partial response in only 1 out of 22 osteosarcoma cases, and no responses were observed in a total of 10 leiomyosarcomas [94]. Nevertheless, some responses to PD-1 blockade have been observed in undifferentiated soft tissue sarcomas, dedifferentiated liposarcomas, myxofibrosarcomas, and chordomas. In the SARC028 trial, for instance, 40% of undifferentiated soft tissue sarcoma and 20% of dedifferentiated liposarcoma patients responded to pembrolizumab. Interestingly, only half of the responsive cases of undifferentiated soft tissue sarcomas expressed PD-L1 [94]. Similarly, a case study involving a metastatic chordoma observed clinical benefit from pembrolizumab treatment. This particular chordoma did not express PD-L1 before treatment but did contain a loss-of-function mutation in *PBRM1*, which again hints at a role for the chromatin-remodeling complex in response to PD-1 blockade in chordomas [95]. The fact that PD-L1 expression is not associated with response to PD-1 blockade highlights its inaptness as a predictive marker in sarcomas. In line with these findings, half of the sporadic cases of sarcomas with MMRd lack PD-L1 expression. However, MMRd sarcomas are still considered eligible for immune checkpoint inhibition as they often harbor a high mutational burden [91,96]. Some interesting and illustrative examples of the importance of TMB for response to checkpoint blockade are UV-induced angiosarcomas and occasional cases of myxofibrosarcomas and (UV-induced) undifferentiated soft tissue sarcomas [6,75,76,97,98,99]. All of these highly mutated tumors responded exceptionally well to anti-PD-1 therapy. In addition, B cells and TLSs could be other predictive markers for response to PD-1 blockade in soft tissue sarcomas [24]. In their innovative study, Petitprez and colleagues categorized the pre-treatment tumors from the SARC028 trial into sarcoma immune classes and found that half of the responsive cases (5 out of 10 patients) had similar gene expression signatures as the immunologically “hot” class enriched with B cells and TLSs. The other responsive cases were observed in the second immunologically “hot” class and in the “vascularized” immune class, comprising mostly endothelial cell expression signatures. This observed association between the TME and response to immunotherapy again underlines the importance of characterizing the immune microenvironment of sarcomas.

Monotherapy with anti-PD-1 antibodies often falls short in sarcomas, but targeting the immune system remains a promising approach. Therefore, clinical trials have been set up with complementary checkpoint inhibitors, and with success. As an example, ipilimumab, which targets CTLA-4 on T cells, showed beneficial responses when combined with nivolumab, an anti-PD-1 antibody, compared to nivolumab mono treatment in various soft tissue sarcomas, including leiomyosarcomas, myxofibrosarcomas, undifferentiated soft tissue sarcomas, and angiosarcomas [100]. Since dual blockade of CTLA-4 and PD-1 induced responses similar to chemotherapy in these soft tissue sarcomas, combinatory immunotherapy is being further explored in the clinic (https://clinicaltrials.gov/ Identifiers: NCT04741438; NCT04480502; NCT02500797. Accessed on 17 August 2021).

### 3.2. Response to Other Immunotherapeutic Agents

In addition to T cell checkpoint blockade, other immunotherapeutic strategies have been explored in sarcoma, although to a lesser extent. These include, for instance, T cell receptor (TCR) gene therapy and cancer vaccines. TCR gene therapy involves the exploitation of a TCR that recognizes a specific HLA/peptide complex and which is transduced into a patient’s own T cells [101]. Once administered, these modified T cells should elicit a robust antitumor immune response when the target antigen is expressed by cancer cells. TCR gene therapies can be particularly useful when a group of tumors expresses a well-defined cancer-associated antigen. Therefore, this strategy is highly interesting for simple genome sarcomas that, in general, lack neoantigens derived from somatic mutations. A well-known example is the targeting of the tumor-associated antigen New York Esophageal Squamous cell carcinoma 1 (NY-ESO-1) in synovial sarcomas, which has been explored in multiple trials [102]. NY-ESO-1 is found to be expressed in 49.3–82% of synovial sarcomas [103]. One trial using TCR gene therapy against a total of 42 NY-ESO-1 expressing synovial sarcomas showed partial responses in 14 patients and a complete response in one patient [104]. In addition to NY-ESO-1, tumor-associated antigen Preferentially Expressed Antigen in Melanoma (PRAME) is highly expressed in synovial sarcomas and was suggested to be a suitable target for TCR gene therapy as well [105]. However, a downside of TCR gene therapy is that T cells can only recognize antigens in complex with HLA class I molecules. The fact that synovial sarcomas, like other less immunogenic tumors, often lack expression of HLA class I might complicate their targeting. Nevertheless, combining lympho-depleting agents with TCR gene therapy has proven successful in overcoming this obstacle and in maintaining antitumor immune responses in synovial sarcomas [104]. Since synovial sarcomas are not the only subtype expressing such tumor-associated antigens, applying TCR gene therapy in immunologically “cold” sarcomas can prove beneficial in the clinical management of these malignancies.

Another path to the generation of antitumor immunity is the exploitation of cancer antigens through vaccination. These can consist of tumor-associated antigens but also neoantigens in a personalized setting [106,107]. Compared to other cancer types, only a handful of vaccines are currently being tested in sarcomas. One vaccine targeting NY-ESO-1 makes use of a lentiviral vector which is preferentially taken up by dendritic cells and subsequently elicits an antitumor T cell response [108]. A phase I trial including 24 sarcoma patients only showed partial response in 1 patient and stable disease in 13 patients [109]. In line with these findings, a sequential phase II trial that explored the combination of this NY-ESO-1 vaccine with anti-PD-L1 treatment in 45 sarcoma patients observed a partial response and stable disease in one and 23 patients, respectively [110]. To further improve T cell activation upon antigen presentation by dendritic cells, combination therapy of TCR gene therapy and vaccination is currently being explored in the clinic (https://clinicaltrials.gov/ Identifier: NCT03450122. Accessed on 17 August 2021). In so-called pulsed dendritic cell vaccines, autologous dendritic cells are pulsed, i.e., loaded, with tumor lysate. Subsequently, the dendritic cell can present the tumor antigens and thereby trigger an immune response. A phase I/II trial using this method has been performed in bone and soft tissue sarcomas. While significantly increased levels of interferon-γ and IL-12 were observed, indicating an increased immune response, only one patient out of 35 showed a partial response to the vaccine [111]. A vaccine against advanced chordoma, in which the tumor-associated antigen brachyury is targeted, has recently finished a phase II trial [112]. Unfortunately, no significant differences were found in the overall response between the treated and control group.

## 4. Cues from the Tumor Microenvironment for the Development of Novel (Immune) Therapeutic Approaches

### 4.1. Modulating the TME of Sarcomas towards Immunologically “Hot”

So far, T cell checkpoint blockade has not delivered sufficient clinical benefit in the majority of sarcomas, largely due to their immunologically “cold” TME. As immune “hot” sarcomas are more likely to respond to checkpoint blockade, modulating the TME of immune “cold” sarcomas towards a pro-inflammatory phenotype could prove advantageous in enhancing the efficacy of checkpoint blockade in sarcomas (Figure 2A). One attractive path towards sensitizing sarcomas to checkpoint blockade immunotherapies is the combination of the latter with traditional chemo- or radiotherapy treatments. For instance, the immune status of the TME of osteosarcomas was shown to be effectively altered by making use of conventional chemotherapy [113]. After neoadjuvant chemotherapy, resected osteosarcomas displayed an increased density of TILs and an increased expression of PD-L1, thus converting to an immunologically “hot” microenvironment. In line with this, a combination treatment of cyclophosphamide and pembrolizumab has been explored in soft tissue sarcomas [114]. However, effective responses were limited to 6% of the cohort. Furthermore, elevated levels of TAM infiltrate together with an upregulation of the inhibitory enzyme indoleamine-2,3-dioxygenase (IDO) were observed, indicating the onset of an immunosuppressive TME [114,115].

In addition, radiotherapy complementary to PD-1 blockade is currently being investigated. As an example, chordomas are generally treated with radiotherapy but often recur. Since these tumors are considered immunogenic, and no clinical trials with mono checkpoint inhibition have yet been finished, anti-PD-1 immunotherapy is being explored in combination with radiotherapy (https://clinicaltrials.gov/ Identifier: NCT02989636. Accessed on 17 August 2021). This is also an interesting combination for those types of soft tissue sarcomas that are often treated with radiotherapy, as radiotherapy was found to affect the TME by inducing the cGAS-STING pathway as a result of the accumulation of cytosolic DNA [116]. This accumulation results in interferon-γ signaling and upregulation of HLA class I and could, therefore, enhance both lymphocytic infiltration and recognition of neoantigens by T cells [68,117]. Stimulating HLA class I expression by adding interferon-γ to immune checkpoint inhibition has been investigated in synovial sarcomas [118]. The interferon-γ treatment did not only modulate the TME towards a more immunogenic phenotype by induction of HLA class I expression and T cell infiltration but also induced PD-L1 expression. Given the immunologically “cold” TME in sarcomas in general, the opportunity to alter the TME towards an immune “hot” state through conventional therapies could provide clinical benefits as an addition to checkpoint blockade.

Besides modulation with conventional therapies, the immune microenvironment of sarcomas may also be directly targeted. More immune checkpoints have been identified in recent years, offering possibilities for dual therapy with anti-PD-1 or anti-PD-L1 antibodies, of which a comprehensive overview is provided in the recent review article by Zhu and colleagues [119]. Such therapeutic targets in sarcomas include T cell Immunoglobulin and Mucin-domain containing-3 (TIM-3), Leukocyte Activation Gene-3 (LAG-3), and T cell Immunoreceptor with Ig and ITIM domains (TIGIT). Both TIM-3 and LAG-3 are expressed on immune cells and are co-expressed with PD-1 within the TME of immunologically “hot” soft tissue sarcomas, whereas T cells and NK cells highly express TIGIT in several osteosarcomas [24,120,121]. Interestingly, most of these “next-generation” immune checkpoints are co-expressed with PD-1, which means that dual therapy with PD-1 blockade is a promising therapeutic option. Indeed, a phase I/IIa trial utilizing co-inhibition of LAG-3 and PD-1 has already shown improved efficacy in melanomas compared to anti-PD-1 monotherapy, and this will soon be investigated in sarcomas as well (https://clinicaltrials.gov/ Identifier: NCT04095208. Accessed on 17 August 2021) [122].

Since the TME of sarcomas often contains large numbers of immunosuppressive TAMs, it makes sense to investigate these immune cells further and find ways to polarize their pro-tumoral anti-inflammatory (M2-like) properties towards antitumoral pro-inflammatory (M1-like) activities. Similar to the aforementioned immune checkpoints, macrophages also have several receptors involved in their regulatory functions, which can be pursued for treatment, such as signal-regulatory protein alpha (SIRPα) and colony-stimulating factor 1 receptor (CSF-1R). SIRPα inhibits the phagocytic activity of macrophages when interacting with its ligand CD47 which can be expressed on tumor cells [123]. In many sarcomas, including chordomas, dedifferentiated liposarcomas, and osteosarcomas, CD47 was observed to be highly expressed on tumor cells along with SIRPα expression on macrophages, suggesting a means of immune evasion in these tumors through this inhibitory axis [39]. Furthermore, macrophages polarize towards an M2-like phenotype upon stimulation of CSF-1R by its ligand CSF-1 [124]. In leiomyosarcomas and osteosarcomas, CSF-1R was found to be highly expressed by TAMs, and, in leiomyosarcomas, expression of CSF-1 and related proteins have also been associated with worse clinical outcomes [125,126]. Interestingly, CSF-1R expression has been found to be associated with good prognosis in osteosarcomas [127], which is in line with the reported protective function of TAMs in osteosarcomas [128]. Since CSF-1R can be expressed by both M1-like and M2-like TAMs, it is still unclear what specific roles these macrophages have in sarcoma genesis. By inhibiting the CD47/SIRPα or the CSF-1/CSF-1R axis, macrophages can be stimulated to exert their phagocytotic function or can be guided towards more pro-inflammatory phenotypes, respectively [129,130]. This can potentially aid in mounting efficient antitumor immune responses as it might sensitize sarcomas for checkpoint blockade. In the coming years, several trials will be held that apply macrophage-targeting therapeutics (https://clinicaltrials.gov/ Identifiers: NCT04751383; NCT04242238. Accessed on 17 August 2021).

### 4.2. Future Prospects in Engineered T Cell Therapy and Cancer Vaccines in Sarcomas

Over the years, other methods next to TCR gene therapy which genetically modifies T cells, have gained interest as clinical applications in sarcomas. For instance, chimeric antigen receptor (CAR) T cell therapy is an approach where patient’s T cells are collected from their peripheral blood and genetically modified ex vivo through the introduction of a CAR [131]. The CAR is composed of a variable fragment of an antibody and a T cell signaling domain. Afterward, the CAR-expressing T cells can recognize tumor-associated antigens that are expressed at the surface of tumor cells, in an HLA-independent context, and mount an immune response against the tumor (Figure 2B) [132]. Ongoing clinical trials involving CAR T cell therapy are directed at a multitude of tumor-associated antigens regularly found in sarcomas, but predominantly in osteosarcomas and Ewing sarcomas. These antigens, or proteins, in this case, include the epidermal growth factor receptor 2 (HER2), disialoganglioside (GD2), and B7 homolog 3 (B7-H3). They have all been found overexpressed in tumor tissue but not in normal tissue [133,134,135]. Although CAR T cells promisingly mediate anti-or immunity in vitro, overcoming the barrier formed by the TME remains difficult in vivo. Especially since the TME of sarcomas generally encompasses immune suppressive cells, finding a way to circumvent these will be necessary to improve the therapeutic efficacy of CAR T cells. Furthermore, it is known that infiltration of CAR T cells into solid tumors is challenging due to physical barriers, e.g., vascular endothelium, and the lack of chemo attractants, thereby limiting the therapy responses [136]. Nonetheless, similar to T cell checkpoint blockade, lymphodepletion (cyclophosphamide or all-trans retinoic acid) or adjuvant cytokine treatment (IL-2, IL-12, or IL-15) can aid in improving CAR T cell-induced immune responses in sarcomas [137,138].

In the context of cancer vaccines, targeting neoantigens originating from gene fusions has become an interesting strategy for translocation-driven sarcomas (Figure 2B). In theory, antigens derived from gene fusions can be more immunogenic than most point mutations due to the joining of two open reading frames. Although not compared to SNVs, it has recently been shown that gene fusion-based neoantigens can indeed stimulate T cell responses [139]. In an older study, the antigenicity of fusion proteins of several sarcoma subtypes has been assessed by the binding ability of peptides from the fusion breakpoints to HLA class I [140]. Two peptides derived from the *SS18-SSX* gene fusion in synovial sarcoma showed specific binding to HLA-A24 molecules. Later, the *SS18-SSX* gene fusion in synovial sarcoma was targeted with vaccines [141,142]. One pilot clinical trial showed increased CTL frequencies after vaccination in nine out of 21 patients. However, shrinkage of the tumor was only observed in one patient. Although the majority of synovial sarcomas do not seem to profit from cancer vaccines targeting the *SS18-SSX* gene fusion, many other neoantigens derived from gene fusions that drive the oncogenesis in sarcoma types have not yet been included in clinical trials. Furthermore, novel gene fusions are still being discovered in sarcomas, indicating a potential for the future [143,144,145].

## 5. Conclusions

The composition of the TME in sarcomas is highly influenced by their genome. Since genetically complex sarcomas are consistently infiltrated by larger quantities of immune cells than sarcomas with simple genomes, it can be postulated that genomic complexity plays an influential role in conferring an immunogenic character to sarcomas. The accumulation of chromosomal alterations, such as copy number alterations and structural variants, generates extensive genetic heterogeneity in complex sarcomas. Consequently, immune-related pathways become deregulated, thereby provoking either pro-inflammatory or immunosuppressive signals which all contribute to diverging compositions and immune statuses of the TME. Because of these heterogeneous features of many sarcomas, understanding the mechanisms that shape their TME remains intricate. However, state-of-the-art tools, such as single-cell approaches, spatial transcriptomics, and proteomics, have enhanced our ability to study the immunophenotypes of the sarcoma TME in detail and could help to overcome these challenges.

So far, immunotherapeutic options for sarcomas have not yet booked great successes. An encouraging strategy for sarcomas with a “cold” is the modulation of their TME towards an inflammatory state. This can be established by, for instance, by employing chemo- or radiotherapy or “next-generation” checkpoint inhibitors, thereby increasing the efficacy of established immunotherapies. Furthermore, T cell engineering therapies and cancer vaccines are highly promising in sarcomas as specific neoantigens, or tumor-associated antigens can be targeted directly. In the coming years, utilizing high-resolution spatial technologies will help identify immune cells in the TME of sarcomas associated with clinical responses, which will aid in making sarcoma patients better amenable to immunotherapy.

Even though most sarcomas are generally considered immune “cold,” there is sufficient evidence of immune infiltration and antitumor immune responses in their microenvironment. Deepening our understanding of the TME can aid in discovering suitable biomarkers and novel targets for immunotherapy and therefore improve clinical management of sarcomas.

## Figures and Tables

**Figure 1 biomedicines-09-01048-f001:**
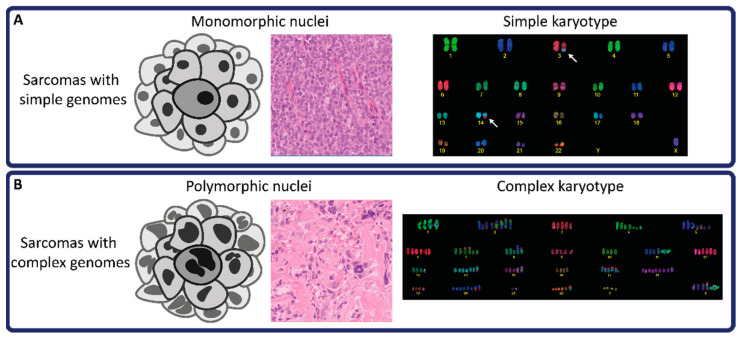
Morphological and karyotypic differences between sarcomas with simple and complex genomes. (**A**) schematic figure and HE staining demonstrating monomorphic nuclei. White arrows in simple karyotype image indicate a translocation (3:14), typically observed in translocation-driven sarcomas. (**B**) schematic figure and HE staining demonstrating polymorphic nuclei. The complex karyotype image displays extensive chromosomal aberrations, including chromosomal gains and translocations.

**Figure 2 biomedicines-09-01048-f002:**
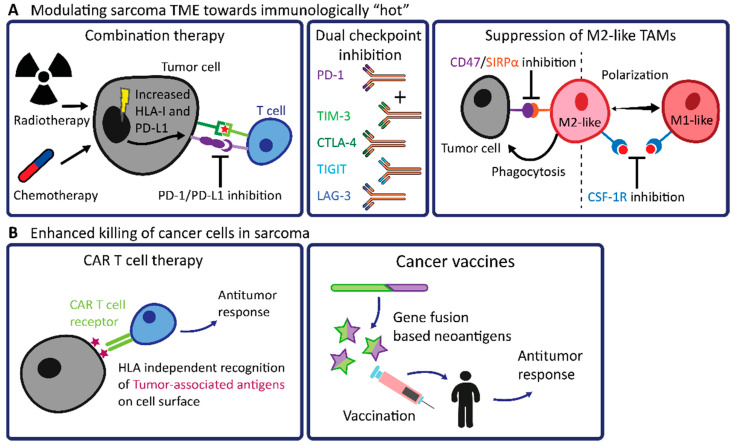
Overview of high-potential future therapeutic approaches for sarcomas with simple genomes or sarcomas with immunologically “cold” or excluded TME.

**Table 1 biomedicines-09-01048-t001:** An overview illustrating TME characteristics of a selection of simple and complex genome sarcoma types.

Sarcoma Type	DNA Alterations	Immune Infiltration ^1^	PD-L1 Expression ^2^	Genomic/TME Heterogeneity ^1^	General TME Composition
***Simple genome***					
Alveolar soft-part sarcoma	*TFE3-ASPCR1*	−	29.7–100%	−/−	CD8 T cells; TAMs [25,26,27]
Chondrosarcoma (low-grade)	*IDH, COL2A1*	−	0%	−/±	CD4 and CD8 T cells; TAMs [28,29,30]
Well-differentiated liposarcoma	*MDM2*, *CDK4*, CNA	−/±	0–50%	−/−	CD4 Th and CD8 T cells; B cells, DCs; TAMs [31,32,33,34]
Ewing sarcoma	*EWSR1-ETS*	−	0%	−/−	M2-like TAMs; CD4 and CD8 T cells [34,35,36,37]
Synovial sarcoma	*SS18-SSX*	−	0%	−/±	TAMs; CD4 FOXP3 Tregs and CD8 T cells [34,35,38]
***Complex genome***					
Chondrosarcoma (dedifferentiated)	*IDH*, *COL2A1*, CNA	±	41–52%	+/±	CD4 and CD8 T cells; TAMs [29,30]
Chordoma	*CDKN2A*, *PBRM1*, *SMARCB1*, CNA	+	0–68.5%	±/±	CD4 FOXP3 Tregs and CD8 T cells; M1-like and M2-like TAMs [39,40,41,42]
Dedifferentiated liposarcoma	*MDM2*, *CDK4*, CNA	+	10–67%	+/+	CD4 Th and CD8 T cells; B cells, DCs; TAMs [24,31,32,33,34]
Myxofibrosarcoma	CIN	+	16–20%	+/+	CD4 Th, CD4 Treg and CD8 T cells; B cells; DCs; M1-like and M2-like TAMs [34,39,43,44,45]
Osteosarcoma	CIN	±	0–25%	+/+	CD4 and CD8 T cells; M1-like and M2-like TAMs, NK cells; DCs [36,46,47,48]
Soft tissue leiomyosarcoma	CNA	±	34–59%	+/+	M2-like TAMs, CD4 T cells [32,39,43,49]
Undifferentiated soft tissue sarcoma	CIN, SNV	+	0–33%	+/+	CD4 Th, CD4 Treg and CD8 T cells; B cells; DCs; M2-like TAMs [24,34,39,43,45,50,51]
Uterine leiomyosarcoma	CNA	±	0–70%	+/+	M2-like TAMs; CD4 T cells; NK cells [39,43,49,52,53]

^1^ Amount of immune infiltration, genomic heterogeneity, and TME heterogeneity: −, low; ±, moderate; +, high. ^2^ Percentage of tumors expressing PD-L1 in tumor cells. Abbreviations: CIN, chromosomal instability; CNA, copy number alteration; DCs, dendritic cells; MDSCs, myeloid-derived suppressor cells; PD-L1, programmed death-ligand 1; SNV, single nucleotide variant; TAMs, tumor-associated macrophages; Th, T helper cells; TME, tumor microenvironment; Tregs, regulatory T cells.

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
