# Peer review of "Linking Immunity with Genomics in Sarcomas: Is Genomic Complexity an Immunogenic Trigger?"

_biomedicines, 2021, doi:10.3390/biomedicines9081048_

Round 1

Reviewer 1 Report

A comprehensive review on the current understanding of tumor microenvironment in sarcomas and its link with immunotherapy response. I have no concerns with the current format. 

Reviewer 2 Report

The authors present an famous overview of sarcomas genes mutation and immunology. The published data are relevant and the text is good for understand of sarcomas biology. 
i have only two remarks for authors:

can we use some expresions of cell cytokins, receptors and ligands or the mutatio of genes use for exact sarcomas type diagnose by imuno histology mehod application?

can we use this results for a therapy protocol modification? 
if yes would be possible added this fact t o conclusions? 

Reviewer 3 Report

van Oost et al. arranged the current knowledge and issues concerning immunotherapy against sarcomas and discussed novel therapeutic strategies to increase clinical efficacy. Therefore, the reviewer thinks it is worth publication. First, however, the reviewer asks for some revision to understand the authors’ opinion more readily.

  1. The reviewer thinks that the Graphical abstract and the overview of the tumor microenvironment (TME) of simple and complex genome sarcoma types (Table 1) are helpful for medical oncologists and clinical researchers. Information in the body text (main part) of the paper is also helpful. However, the authors did not use their graphical abstract and Table 1 for their explanations in the body text—especially, there seems to be no relation between section 2 and Table 1. The reviewer asks to use Table 1 for the description in the section.

  1. The authors described clinical responses to immunotherapy and novel immune therapeutic approaches in the separated sections. However, they are in the same column in the graphical abstract. Is it possible to show them in order that the reading audience can distinguish them?

  1. The image of chromosomal instability in the graphical abstract may be hard to understand for non-experts. Therefore, the reviewer recommends adding some detailed graphical images of sarcomas with both simple and complex genomes in section 2. Also, add differences between immune cold and immune excluded in the body text in section 2.

  1. Is it possible to add graphical images for the explanation of sections 3 and 4? Unfortunately, the description of the right column of the graphic abstract is too elliptic.

  1. (TME) should be added after the tumor microenvironment at line 93.
